# Macroevolutionary brain scaling is a microevolutionary metaphenomenon

Joanna Baker [1,2] ✉, Robert A. Barton [3] & Chris Venditti [2] ✉

From bees to blue whales, it has long been assumed that brain size scales with body size according to a simple log-linear relationship – with differences in the slope and intercept observed amongst different groups of animals. However, recent analyses in mammals contradict this view, revealing size dependency in the form of curvature in the brain and body mass relationship. Here, we use data from 4679 species across seven animal classes and spanning nearly 12 orders of magnitude to uncover near universal curvilinearity. We demonstrate that this body size dependence is a metaphenomenon emerging from a pattern of diminishing allometry within species with increasing body mass. This has fundamental implications for how we interpret macroevolutionary patterns – which can arise as a consequence of within-lineage dynamics. Our integration of inter- and intra-specific allometries reshapes perspectives on morphological evolution by providing a broader framework for understanding how microevolutionary within-species dynamics shape macroevolutionary phenomena.

From bees to blue whales, how and why variation in brain size among animal species arose has received a lot of attention over the years e.g.[1–6]. It is widely recognized that the strong correlation with body size is a key consideration in understanding macroevolutionary patterns in brain size. For nearly a century, the relationship between brain and body size has been almost universally described as a simple log-linear relationship with variability in slope and intercept amongst major clades, taxonomic or otherwise[3,4,7]. The entrenched assumption of log-linearity underpins all contemporary empirical research and theoretical models seeking to understand brain size evolution as well as that of many other traits. However, there are indications that some biological traits scale with body size on a log-curvilinear scale such as metabolic rate[8–11], ingestion rate[12], and offspring size and number[13]. Recent evidence has found that log-curvilinearity also characterizes the relationship between brain and body size amongst mammals and birds[5], with profound implications for studying relative brain size. A macro-evolutionary size-dependency in the brain and body size relationship is a significant finding that resolves several long-standing puzzles surrounding brain size evolution in mammals, including the taxon-level effect, apparent brain to body size lag effects, and apparent differences in scaling parameters[7,14–16]. These artefacts of fitting linear models are size-dependent phenomena:

complex statistical models and elaborate biological explanations are obviated by fitting a simple quadratic relationship. Two outstanding questions arise, however: how widespread is this pattern across taxonomic groups and what is its underlying cause?

Here, we demonstrate an unprecedented and striking size-dependency that holds across the animal tree of life. Using phylogenetic comparative approaches and a comprehensive brain size dataset spanning 4679 species across nine major animal classes, we find a single curvilinear relationship between brain and body size. Through hierarchical modelling incorporating individual-level variation, we then demonstrate that this curvature emerges as a metaphenomenon driven by diminishing allometry within individual species as body size increases. This previously overlooked size dependency reveals a mechanistic bridge between within-species dynamics and large-scale macroevolutionary patterns.

## Results and discussion
### A near universal size-dependency in the brain-body mass relationship

Using a dataset spanning nine major groups of animals (mammals, birds, testudines, squamates, crocodilians, amphibians, bony fish,

[1]School of Life, Health and Chemical Sciences, The Open University, Walton Hall, Milton Keynes, UK. [2]School of Biological Sciences, University of Reading, Reading, UK. [3]Department of Anthropology, Durham University, Durham, UK. ✉e-mail: joanna.baker@open.ac.uk; c.d.venditti@reading.ac.uk

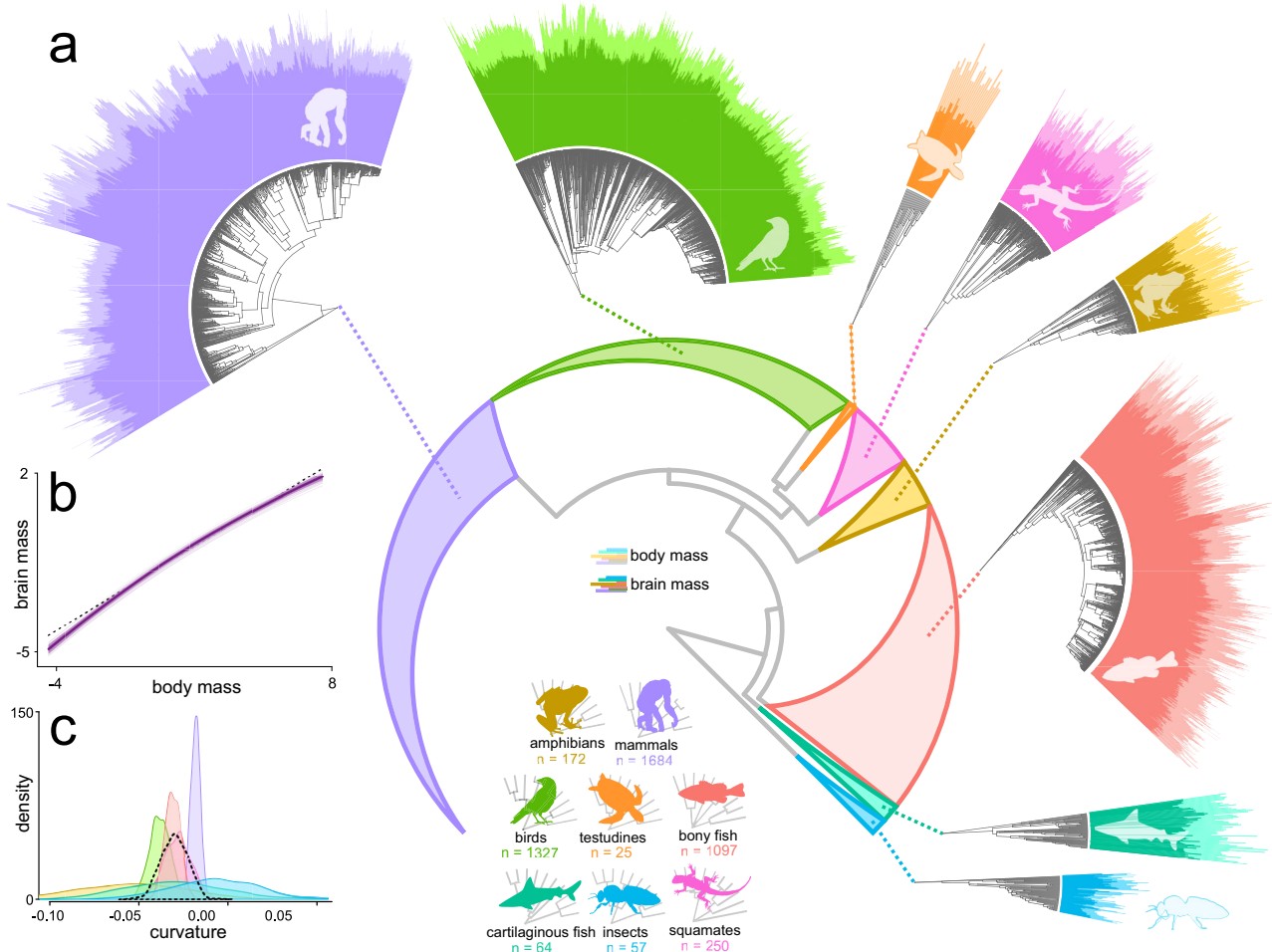

**Fig. 1 | Brain and body size across 4679 animal species. a** The eight major animal classes (those listed above, excluding crocodilians owing to their small sample size) are shown in the central phylogeny with clades collapsed to triangles sized proportionally to the number of species. Each clade is expanded to show finer phylogenetic structure, with bars at the tips proportional to body size (pale) and brain size (dark). **b** The predicted relationships estimated in our global curve model across all 4679 species with the median prediction superimposed upon a random sample ($n = 100$) of the posterior. The median prediction from our global slope model is shown for comparison with a dashed line. **c** The posterior distribution of estimated curvature (quadratic parameters) for each of the eight non-crocodilian classes ($n = 4676$) compared to the global (grand mean) curvature outlined with a black dashed line. All groups substantially overlap the grand mean and are thus not significantly different from a global curve estimated across all species. Silhouettes are not proportional to size and are shown for illustrative purposes only. The major animal clades are indicated by the coloured silhouettes; the colour scheme is used throughout the manuscript to facilitate quick comparisons between results.

cartilaginous fish, and insects) and nearly 12 orders of magnitude in brain and body size (Fig. 1a), we seek to determine whether curvilinearity in the brain and body mass (BBM) relationship is ubiquitous or even universal across the tree of life. To do this, we use a phylogenetic comparative approach, the variable rates regression model[17,18], to study the BBM relationship across a comprehensive time-scaled phylogenetic tree encompassing all species in the sample ($n = 4679$). This model allows us to simultaneously estimate the slope (and/or curvature) of the BBM relationship alongside rate heterogeneity – using a Bayesian reversible jump MCMC procedure to automatically identify lineages of the phylogenetic tree in which relative brain size has evolved more rapidly or more slowly relative to body size.

We first estimated a global curve model across all species. Across all of the major animal groups listed above (henceforth referred to as classes for simplicity), there is an overall curvature in the BBM comparable to that found in mammals[5], with a significantly negative quadratic parameter (median = −0.014, $p_x = 0.000$) and a positive slope (median = 0.614, $p_x = 0.000$). The global curve model is significantly supported (in terms of Bayes Factors[19], BF, see methods) over both a global slope model estimating only a single slope and intercept across all taxa (BF = 70.52) as well as a class slope model

estimating a separate intercept and slope for each class, considering non-avian reptiles as a single group (BF = 69.28). The global curve relationship is depicted in Fig. 1b. We then further test the possibility that mammals drive the observed relationship and ascertain whether any class displays fundamental differences in the BBM relationship – as recently suggested for bony fish owing to their indeterminate growth[20,21]. To do this, we ran an additional model estimating a separate quadratic curve for each class (class curve model, $N = 4679$, see methods). In this model, we find that no class shows any significant departure from the grand mean, i.e. a global curvature estimated across all species (using deviation contrast coding, all $p_x > 0.05$). In terms of Bayes Factors, there is no improvement to be gained from estimating separate curves for each class (BF = 0.112). Repeating this analysis on a dataset removing crocodilians ($N = 3$) and treating squamates and testudines as separate classes reaches identical conclusions – with no individual class demonstrating curvature that differs from the global curvature ($N = 4676$; see Fig. 1c). From these results, we can infer a surprisingly invariant general scaling rule across diverse animal classes.

To further investigate the scale of this phenomenon, we ran an additional subclade curve model in which we estimated a separate

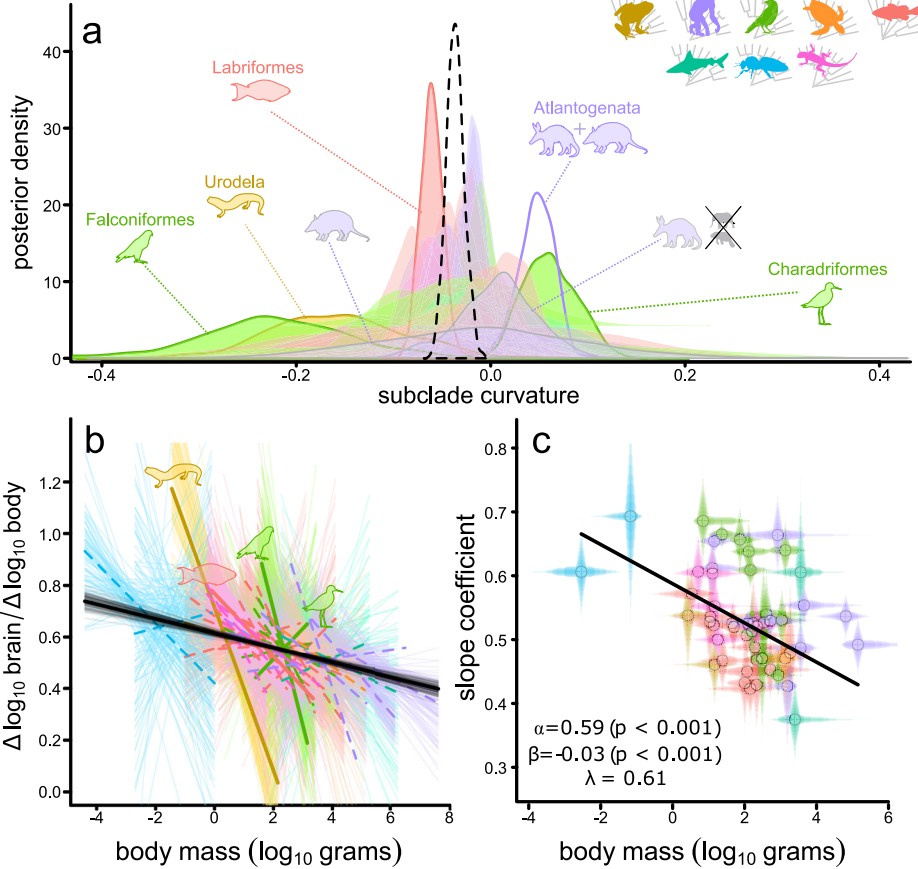

**Fig. 2 | Curvature in the brain and body size relationship across animals. a** The posterior distribution of the estimated global quadratic parameter – dashed black line –observed across $n = 51$ subclades. The posterior distribution of quadratic parameters for each subclade are shown in the colour of the class they belong to and are outlined where they diverge from the grand mean (and indicated by corresponding silhouettes). Atlantogenata (consisting of both Afrotheria and Xenarthra together) substantially differs from the grand mean (purple outline, no fill) – but neither of the subgroups differ after removing sirenians and elephants (grey-outlined purple distributions). **b** The expected change in relative brain size $\left( \frac{\Delta \log_{10} brain}{\Delta \log_{10} body} \right)$, calculated over the full range of observed body sizes for each subclade (separating Atlantogenata into Xenartha and Afrotheria excluding elephants and sirenians) for a random 100 samples of the posterior distribution, with a median estimate overlaid. Lines are solid for the four subgroups which show significant deviations after considering potential outliers and sub-groupings. **c** A negative relationship between linear slope parameters (from a subclade slope model) and body size for animal subclades, obtained using PGLS regression, is significant (two-tailed test). Percentiles of the posterior distribution of slope parameters and body mass ranges are shown as transparent lines for each group. The animal classes are indicated by the coloured silhouettes; the colour scheme is used throughout the manuscript to facilitate quick comparisons between results.

quadratic effect for each of fifty-one monophyletic taxonomically derived subclades with N > = 20 (mostly represented by orders and families, see Materials & Methods). These models are restricted to taxa to which we could reliably assign to subclades ($N = 4457$); there is still significant global curvature in this subset of data. As with our class curve model, there is extraordinarily little departure from global curvature in any subclade. We find significant deviation in only 5 subclades which each only represent a small portion of the range of brain and body size data being studied (Fig. 2A-B). Despite a potentially steeper curve (i.e., a more negative quadratic parameter), three of these groups – Urodela (salamanders), Falconiformes (falcons), and Labriformes (parrotfish/wrasses) – show qualitatively the same pattern of negative curvature as the global BBM relationship, even after considering potential outliers and differences amongst monophyletic groupings within each subclade (Figs. S1–S6).

The other two groups, Atlantogenata (African mammals, sloths, and armadillos) and Charadriiformes (shorebirds) show positive curvature. However, within Atlantogenata, when elephants and sirenians – members of relatively disparate orders separated by millions of years of evolution[22,23] – are removed from the analysis, there is no longer any deviation from the grand mean (see Supplementary Note 1,

Figs. S4 and S6). Amongst Charadriiformes, the positive curvature seems to largely be driven by the sub-clade Charadrii ($N = 35$, Supplementary Note 1, Figs S5 and S6), a group mostly represented by plovers. This clade has been noted to have unusually high sexual dimorphism[24] and sexual selection has been suggested to influence brain size amongst Charadriiformes as a whole[25]. Whether this is the cause of why plovers seem to deviate from the expectations derived from all other animal species is an intriguing question that warrants further investigation – along with why elephants and sirenians appear to be outliers, and why some groups seem to have steeper curves.

However, even considering these few deviations (Figs. S1–S6), our results reveal a remarkable ubiquity of curvature in the BBM relationship across all animal life. The same curvature is identified to fit over most animals included in our study – and is therefore not simply a product of fitting a quadratic term over a dataset spanning nearly twelve orders of magnitude. Whilst the parameter of curvature itself may seem only small (median quadratic coefficient = −0.014), not only is it significant (Supplementary Note 2), but it can also result in important implications for our interpretation of brain size scaling. For example, if we were to ignore the observed curvature in the BBM, a sperm whale (*Physeter catodon*) is predicted to have a brain size of

~12 kg – much too large compared to observed values (~7.8 kg). However, curvilinearity substantially improves this prediction – reducing it more than 43% to a value of ~7 kg and far closer to reality. These results thus have crucial implications for understanding the brain size relative to body size of individual species. In order to visualize how this curvature can be interpreted across different taxa, we calculated the expected change in brain and body size ($\frac{\Delta\log_{10} brain}{\Delta\log_{10} body}$) over the full sample of estimated parameters for each of the subgroups in our analysis (Fig. 2B). If there was no curvature, the expected change in brain and body size would be equivalent for animals of all sizes – yet we see that on a log-scale that large-bodied animals of most subgroups have proportionally less change in brain mass per unit body mass.

In all analyses reported here, we find substantial support for variable rates (see methods, Fig. S7 and Supplementary Note 3). However, the curvilinearity that we identify is supported in the absence of rate heterogeneity. Overall, our results remain qualitatively unchanged when we use a model that assumes a constant rate of evolution. Confirming this, we find further support for a consistent curvature in the BBM relationship in the form of a significant negative association between the slope of the BBM relationship from the sub-clade slope model and the average body size across all fifty-one sub-clades (Fig. 2C; $\alpha = 0.590$, $\beta = -0.031$, all $p$-values < 0.001). This relationship still persists to the exclusion of the small-bodied insects ($\alpha = 0.591$, $\beta = -0.034$, all $p$-values < 0.001) as well as sharks and testudines (the two other groups with relatively small sample size, Fig S8; $\alpha = 0.590$, $\beta = -0.034$, all $p$-values < 0.001); see Supplementary Note 4 for more details. It is also upheld in the absence of rate heterogeneity (Supplementary Note 5). Together, this universal body size dependency underpins the fascinating prospect that heterogeneity in slopes previously attributed to diverse selection pressures or scaling rules can be explained simply as a size-dependent effect[5]. This obviates the need for special explanations for distinct patterns in diverse taxa and has major implications for understanding brain evolution. For example, this moves us beyond the concept of testing for lineage-specific scaling patterns and suggests that some previously identified correlates of relative brain size might be size-dependent artefacts. Across all animals – from bees to blue whales - there is a size-dependency in the BBM (Fig. 2B) that requires an explanation.

## Curvature as a macroevolutionary metaphenomenon

Comparative phylogenetic approaches to studying evolutionary relationships – including those we carry out here – seek to identify a so-called "evolutionary regression coefficient"[26] that describes the covariation of two or more traits along the branches of a phylogenetic tree (i.e., an evolutionary slope parameter). The interpretation of such parameters is implicitly linked to underlying within-species variation[26,27] – whether this arises through measurement error or observational variability including population differences[26–28]. That is, the patterns we are observing at the macro-evolutionary scale are essentially averages of those acting across individual lineages[26] and thus on the variability upon which natural selection acts. This view is reflected in earlier studies of metabolic rates which linked macro-evolutionary metabolic scaling to variation in the intercepts amongst species[29]. However, in the context of the BBM relationship, a curvi-linear model implies that the evolutionary regression coefficient (slope) along individual branches is itself linked to body size in a biased fashion. In such a scenario, variation in species-level brain allometries would be linked to body size. That is, we would predict not just simple variability in intercepts, but an explicit bias wherein within-species slopes will be lower at larger body mass (Fig. 3 and Supplementary Note 6).

Here, we investigated the possibility that heterogeneity in the microevolutionary allometry observed within individual species[4] shows a directional association with body size in such a way that it gives rise to a macroevolutionary metaphenomenon of curvature

(Fig. 3). To do this, we used brain and body size data from Tsuboi et al.[4], spanning a total of 376 vertebrate species with at least 10 individual measurements– note that this dataset does not include any insects or cartilaginous fish. Consistent with our hypothesis, using phylogenetic generalized least squares models[30], we identify a strongly significant negative relationship between the within-species slopes of the BBM relationship and the average body size of each species (Fig. 4a, $\alpha = 0.37$ [$p < 0.001$]; $\beta = -0.04$ [$p = 0.008$]; $\lambda = 0.13$). This is not owing to a reduction in the strength of association in larger species: there is no association of body size with the significance, mean-squared error, or $R^2$ value (Supplementary Note 7) – that is, there is truly a body size dependency in these within-species allometries.

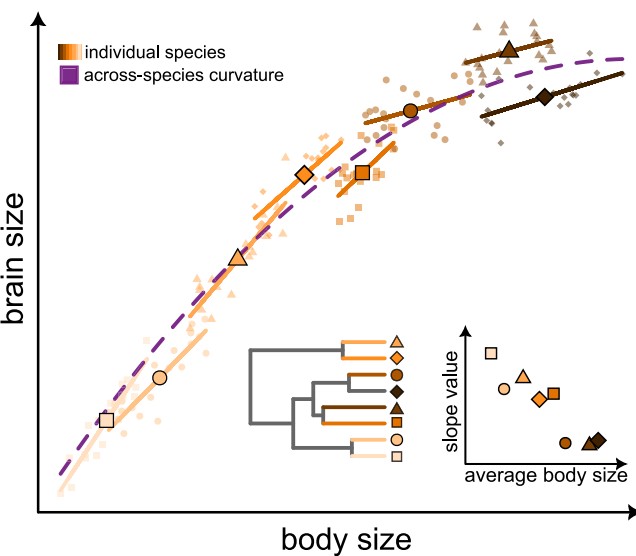

**Fig. 3 | How microevolutionary allometry can give rise to macroevolutionary metaphenomena.** A curved relationship between brain and body size across species can arise from differential allometries observed within individual taxa. If species-level allometries vary with size (inset), this results in a general tendency for the slope in the BBM relationship to become shallower with increasing size (orange lines). When we study the data across species, this gives rise to a macroevolutionary metaphenomenon of curvature (purple dashed line).

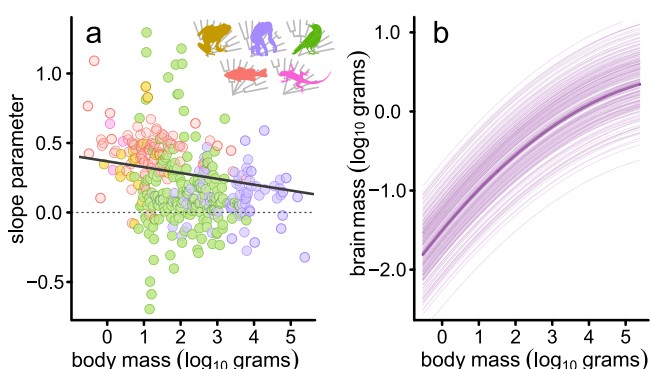

**Fig. 4 | Size bias in microevolutionary brain allometry and figure size bias in microevolutionary brain allometry and across-species curvature is observed in real biological data.** The patterns depicted in Fig. 3 are reflected in real data from 376 vertebrates. **a** There is a significant negative relationship between slope parameter and average species-level body size. **b** There is significant curvature in the relationship across species – shown as the predictions from a global curve model ($N = 376$) accounting for within-species variability in both brain and body size as a random effect in a PGLMM. The median prediction is overlaid on top of a random sample of 100 lines from the posterior. The major animal clades are indicated by the coloured silhouettes; the colour scheme is used throughout the manuscript to facilitate quick comparisons between results.

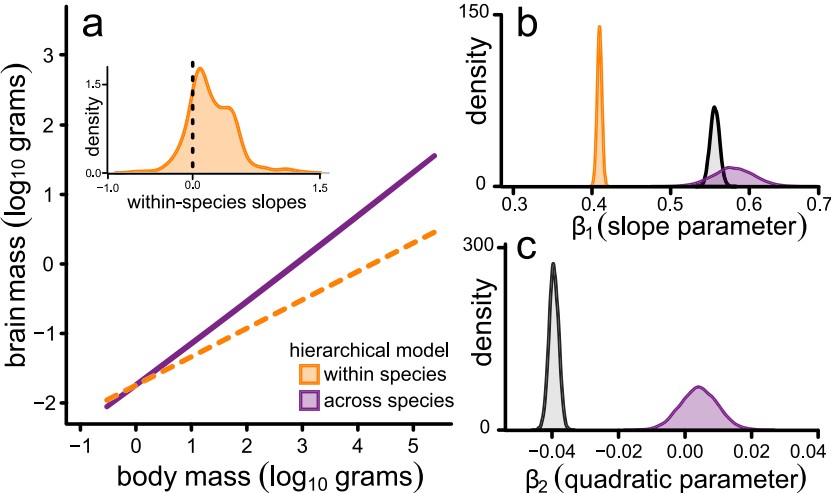

Fig. 5 | Within-species variation in brain allometry explains the macroevolutionary metaphenomenon of curvature in the BBM relationship. a The predicted relationship between brain and body size from the hierarchical model separated into within-species effects (orange) and across-species effects (purple). The average within-species slope is derived from the species-level variability in slope depicted in (b, inset). In (b, c), we plot the posterior distributions of estimated parameter estimates from our hierarchical model. We also include the posterior distribution of parameter estimates from the across species global curve model (from Fig. 4b) for comparison (grey).

Whilst it is recognized that within-species allometries tend to be shallower than those observed across higher taxa[4], we note that many of the slopes within individual species are close to zero – with some of these even being negative. It is difficult – though not impossible[31] – to envisage a scenario in which an increase in body size would lead to a reduction in brain size and so the relationships within these species may warrant further investigation. However, our results stand even when we study only those species with positive slopes ($N = 313$, $\alpha = 0.43$ [$p < 0.001$]; $\beta = -0.05$ [$p < 0.001$]; $\lambda = 0.12$, $R^2 = 0.04$).

In order to distinguish the nature of individual-level variability from macroevolutionary patterns and thus formalise the notion presented in Fig. 3, we can use a statistical technique known as within-group centring[32]. This approach accounts not only for variation in trait values, but also the heterogeneity observed amongst individual species-level allometries. However, before being able to implement these approaches, we first ensured that the curvature was still apparent in the subset of data for which we have individual-level variation ($N = 376$). Using phylogenetic generalized linear mixed models[33] (PGLMMs) to account for individual-level variation in both brain and body size, we recover a significant negative quadratic parameter across species (median = $-0.039$, $p_x = 0.000$) as well as a significant positive slope (median = $0.55$, $p_x = 0.000$) (Fig. 4b). That is, we can explicitly demonstrate the expectations laid out in Fig. 3 using empirical data.

We therefore proceeded to run a PGLMM model using within-group centring[32] – referred to as our *hierarchical model*. Our hierarchical model estimates two main components: (i) a BBM relationship across species including curvature (using species-level means); and (ii) an average effect of within-species allometric variability (using species-level slopes as a random effect). In this hierarchical model, we no longer find any curvature across species (mean $\beta = 0.004$, $p_x = 0.475$), indicating that intraspecific patterns of brain size variation are the driving force behind the macroevolutionary curvature (Figs. 3–4). The relationship acting within individual species is generally much shallower (mean $\beta = 0.41$, $p_x = 0.000$) than that observed across species (mean $\beta = 0.59$, $p_x = 0.000$) as depicted in Fig. 5 a,b. Whilst this is in line with early observations of intraspecific vs. interspecific brain allometry amongst mammals[34], it is important to note that this within-species slope is derived from the true

variability in slope observed within individual species (Fig. 5a, inset). An alternative way of parameterizing this model – with identical interpretation – would be to estimate curvature in the within-species component of the model rather than allowing intra-specific variability (i.e. not including species-level slopes as a random effect). This model, shown in Fig. 6, still removes the across-species curvature while the significant within-species negative quadratic effect (mean $\beta = -0.02$, $p_x = 0.000$) again emphasizes the general tendency for a reduction in slope values for individual species with increasing size – as shown in Fig. 3. Previous suggestions that there may be a low correlation between brain and body size within individual species e.g.[35–37] may be associated with the fact that the most well-studied taxa tend to be larger. In any case, the factors driving such body size dependency as well as additional variability in the within-species BBM relationship remains to be fully elucidated.

Widespread curvilinearity in metabolic scaling e.g.[8], has previously been linked to differences amongst scaling-exponents at lower taxonomic levels[10,38,39]. Along with previous evidence from mammals and birds[5], our results demonstrate similarly widespread curvilinear scaling for brain size – but go one step further by explicitly examining the effects of both within-species and across-species relationships with body size. In doing so, we reveal that it is only by understanding the relationship within individual species that we can actually begin to explain the observed curvature across species. The factors driving variation within species, including genetic factors, fundamentally differ from processes shaping evolutionary diversity which often involve associations with other traits or environmental factors[40–42]. In line with this, several factors commonly reported to be important brain size correlates at the species level have no apparent effect on the size-dependency caused by variability amongst individuals. Variation in neither metabolic rates[43,44] nor neuron number[45,46] affect the curvature across mammals[5], and neither environmental temperature[47] nor diet[48] have any effect on size-dependency amongst endothermic vertebrates (see Supplementary Note 8). Instead, our results imply a mechanistic explanation in which the global curvature in the animal BBM relationship is a manifestation of the change in brain and body size within individual species along a body size gradient. Simply put, within-species variation explains the size dependency in the BBM relationship.

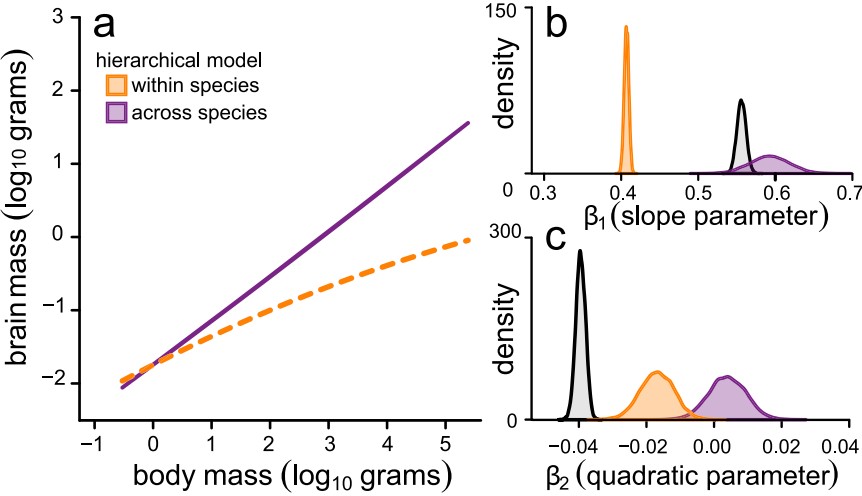

**Fig. 6 | Curvature in the average brain allometry observed within species explains the macroevolutionary metaphenomenon of curvature in the BBM relationship. a** The predicted relationship between brain and body size from an alternative hierarchical model separated into within-species effects (orange) and across-species effects (purple). In this model, we do not incorporate a random effect for species-level slopes (as we did for our main hierarchical model depicted in Fig. 5). The overall curvature in the average within-species slopes is owing to the pattern of diminishing slope value with increasing body size. In (**b**, **c**), we plot the posterior distributions of estimated parameter estimates from our alternative hierarchical model. We also include the posterior distribution of parameter estimates from the across species global curve model (from Fig. 4b) for comparison (grey). In this alternative hierarchical model, there is no significant curvature across species.

## Concluding remarks

Our results provide unprecedented insight into what exactly it is we are studying when we use phylogenetic regression. By studying traits such as relative brain size at multiple scales – from individuals to species – we can reveal profoundly different patterns and provide crucial insights into how we measure relative brain size – and how macroevolutionary patterns can arise from within-species variation. For example, Puschel et al.[35] recently revealed that hominin encephalization resulted from increases in brain size within individual species and an accelerating pattern of increase through time. Whilst an increase in the rate of brain expansion during hominin evolution has been repeatedly demonstrated[49–51], it is only by the separation of within- and between- species evolutionary dynamics that we can truly begin to understand how these patterns have arisen. It may not always be enough to simply study species-level averages whilst accounting for variation in trait values, i.e. Figure 6a, refs. [52–54]. Instead, we need to understand how differences in the BBM arise through evolutionary processes acting at different hierarchical levels[4,32]. That is, there is a necessity to understand how microevolutionary variation can give rise to macroevolutionary patterns.

The universal curvilinearity in the brain and body size relationship observed across species is a metaphenomenon driven by microevolutionary within-species processes. This raises questions about studying brain size evolution in relation to behavioural-ecological factors – wherein incorporating body size dependency or within-species dynamics may affect our conclusions about the drivers of brain evolution. Furthermore, this size dependent phenomenon is unlikely to be unique to brain size.

It is already well-established that non-linearity and curvature exists in the relationship between metabolic rate and body size across a wide range of organisms[8–12] and authors have noted a potential size-dependent effect[11] similar to that observed here for animal slopes. Curvature in fundamental biological traits has profound implications for the interpretation of many biological phenomena such as biological scaling[55–57]. For instance – and given the importance of metabolism in almost all biological variation and processes[58,59] – it follows that we should perceive curvature in other biological traits[12]. Indeed, along with brain size, non-linearity has been demonstrated in a suite of other traits including, ingestion rate[12], locomotion costs[12], maternal energy intake[60], and offspring size and number[13]. Our results for brain size indicate that we need to look to variation within individual species in order to be able to explain these patterns. This is in line with observations made for metabolic rates in which curvilinearity can be attributed – at least to some degree – to ontogenetic variation within individual species as well as to interspecific relationships[55,56].

To the extent that such patterns are true, our approach reveals a way to more clearly understand the underlying causes of enduring and controversial macroevolutionary phenomena such as Cope's rule[61] and Bergmann's rule[62] - using a simple separation of microevolutionary and macroevolutionary processes. A deeper understanding of what exactly we are studying when we use phylogenetic regression provides a broader framework for understanding how microevolutionary within-species dynamics can shape macroevolutionary phenomena.

## Methods

### Species-Level Data

The brain and body sizes of animal species were collated from the literature. The full dataset and all references are provided as Supplementary Data 1. We chose to use only whole-brain mass and excluded species where brain masses were reported excluding the olfactory bulb e.g.[63,64]. Where measurements were explicitly noted to come from juvenile specimens e.g.[4], we did not include these measurements. We preferred sources which provided paired brain and body size estimates, and those which reported brain size as masses rather than volumes. Where only endocranial volumes were available, we converted these to masses in line with a $1\,g = 1\,cm^3$ conversion, ensuring consistency with conversions used in several major sources[3,65,66]. However, the pattern is upheld even in groups where data is primarily derived directly from masses (e.g. Chiroptera, Amphibians, cartilaginous fish). Where duplicate entries occurred for species given these criteria, we preferred the earliest source after removing extreme outliers by eye. In all cases, we record the source from which we obtained the data used in our analysis in Supplementary Data 1.

In some cases, we modified values or used values from an alternative source. We excluded the brain mass for the blue whale *Balaenoptera musculus* in several sources owing to a misrepresentation

derived from a dehydrated sample[67]. Rodentia and Lagomorpha brain sizes from several sources[2,4,68] were adjusted as outlined by[66,69] in order to correct a systematic bias in the original source from which they were obtained[70]. For fish and sharks obtained from the BRAINS table in Fishbase[71], we used the average reported brain and body size per species (although analyses using the maximum made no difference).

Our final dataset contained brain and body size data for a total of 4679 species that were also found in the animal-wide phylogenetic tree (see below). The full dataset is provided as Supplementary Data 1. All data were logged (log$_{10}$) before analysis.

## Phylogenetic Tree
For all our phylogenetic analyses, we use the time tree of life[72–74]. We downloaded a phylogenetic tree for Chordata from the time tree of life website in February 2024[75]. Species names were matched to the time tree; only species found in the tree were retained for analysis. We ensured the tree was ultrametric (all tips terminated at the present) and collapsed all polytomies to multifurcations (also referred to as hard polytomies). Finally, owing to numerical issues associated with estimating evolutionary parameters along trees with very short terminal branches, we removed randomly all but one member of any species group descending from a single node with a branch length of less than 0.25 (250 Ky). To do this, we used a custom R function, which we provide as Supplementary Code 1.

## Class-level and subclade assignments
We separated species into major phylogenetic and taxonomic clades: amphibians, mammals, bony fish, cartilaginous fish, reptiles, and birds. As reptiles comprise several major taxonomic groups: birds ($N = 1327$), squamates ($N = 251$), testudines ($N = 25$), and crocodilians ($N = 3$), we divided these taxa into two major monophyletic clades: birds and non-avian reptiles. Our analyses are conducted both considering non-avian reptiles and birds separately (in which the phylogenetic structure amongst the different reptilian groups are incorporated) and additionally studying testudines and squamates as separate groups. In the latter, crocodilians are removed owing to their small independent sample size. All major animal clades are referred to as classes for simplicity and consistency.

We then assigned species to a lower-level taxonomic designation (henceforth subclade) based on major taxonomic resources[76–81] and original references (see Supplementary Data 1). For the most part, these subclades are represented by taxonomic orders. In some cases, subclades were paraphyletic or too small for further study and so to maximize species inclusion, we combined subclades to form larger monophyletic groups. For example, ten different groups of fish (a mix of order and family level) as defined by the NCBI taxonomy database[77] form a single monophyletic clade (Ovalentaria) in the time tree of life and so were here included as a single group. We studied marsupials as a single group as it included several small orders which would be otherwise excluded and similarly studied Afrotheria and Xenarthra within mammals as a single magnorder, Atlantogenata (though see results and Supplementary Note 1 for more information about the division of this group). The major bird clade Aequorlitornithes as described by Prum et al.[81] comprises two large monophyletic groups which we here study as two separate subclades: the shorebird order Charadriiformes ($N = 140$), and another clade containing various other waterbird orders such as loons, penguins, herons, and seabirds ($N = 66$). We retained only subclades large enough for further study (N ≥ 20) – with one exception. We include bees ($N = 14$) in order to enable comparison between insect subclades (bees and ants). $N = 220$ species belonging to all other smaller groups were removed from our subclade-level analyses for a total of $N = 4457$ species across 51 subclades.

## Within-Species Data
The brain and body sizes for individual vertebrates were obtained from Tsuboi et al.[4,82]. Using code associated with and described in the original paper[4,82], we centred data according to sex (male, female, or unreported) and measurement method (volume or mass). We then linked the names in this dataset to the names in the time tree of life and removed species where N < 10, resulting in a dataset of 376 species. Results using uncentred data produced the same major conclusions (Supplementary Note 9).

## Analysis
**The brain and body size relationship across species.** We performed a series of phylogenetic generalized least squares (PGLS) analyses to model the relationship between brain and body size across animal species. Here, we estimate curvature using a quadratic parameter – which is likely to be a good approximation of the curvilinearity observed over the range of observed body masses[5]. Note also that this would essentially be identical in interpretation to a linear threshold model that estimates different slopes for different size classes – with infinite thresholds. Our global curve model estimated a single intercept, slope, and quadratic parameter across all $N = 4679$ species. Our class curve models estimated a separate intercept, slope, and quadratic parameter for each of the major animal classes and were conducted both across all $N = 4679$ species (considering all non-avian reptiles as a single group) and across $N = 4676$ species (considering each reptile clade separately, to the exclusion of crocodilians). Our subclade curve models estimated a separate intercept, slope, and quadratic parameter for each of 51 subclades and is performed on only a subset of $N = 4457$ species (see above). Repeating the class curve model and global curve model on the limited dataset resulted in only negligible differences and so we report the full results here.

All analyses were performed using a Bayesian reversible jump MCMC procedure implemented in BayesTraits[83]. Chains were run for a total of 10 billion iterations, removing the first 100 million as burn-in and retaining every 500 thousand samples. We used a wide and uninformative normal prior centred on zero with a standard deviation of 2.5 for all regression parameters. Convergence was assessed visually, and multiple replicates were run to ensure repeatability of results and that our posterior distributions differed from the prior. Our results did not differ across replicates. We assessed parameter significance using the proportion of the posterior distribution overlapping zero ($p_x$). Where the posterior distribution of a parameter value had $p_x < 0.05$, it means that less than 5% of the distribution crosses zero and could therefore be considered substantially different from zero. We compared differences amongst classes and subclades using sum contrast (or deviation) coding – a coding system for categorical variables in regression analyses that allows for the comparison of a group-level parameter (e.g. class-level curvature) to an overall mean across all species (grand mean curvature). This method returns parameter estimates interpretable as the difference between each group and the grand mean. Therefore, where the posterior distribution of difference parameters significantly overlaps zero ($p_x > 0.05$), there is no substantial difference between that group and the grand mean.

We implement our PGLS models using the variable rates regression model, which simultaneously estimates the regression parameters alongside heterogeneity in the rate of evolutionary change[17,18]. This method automatically identifies lineages (branches) of the phylogenetic tree in which relative brain size has evolved at a faster or slower rate relative to body size[17,18]. It does this by estimating a set of rate scalars $r$ defining the rate of evolution on each branch of the phylogenetic tree: where $r > 1$, a branch is evolving faster than the background rate of change, and where $0 \leq r < 1$, it is evolving slower.

We assessed whether rate heterogeneity was supported in each of our models by calculating Bayes Factors with the equation BF = $-2\log_e(m_I/m_O)$, where $m_O$ is the marginal likelihood of a model with

only a single underlying rate of evolutionary change and $m_1$ is the marginal likelihood of a variable rates model that estimates rate heterogeneity. Where BF > 2, it is considered to be positive support for rate heterogeneity[19]. All marginal likelihoods were estimated using stepping-stone sampling[84], sampling 100,000 iterations for each of 1000 stones after convergence was reached. All PGLS analyses reported in the main text show substantial support for variable rates – all having Bayes Factors > 500. A summary of the estimated rate for each branch of the phylogenetic tree is provided in Supplementary Data 2. However, it is not possible to estimate rate heterogeneity in all downstream analyses (e.g. mixed models, see below). Therefore, we also replicated all our PGLS analyses using models that assumed a constant rate of evolution (i.e. no rate heterogeneity). Our conclusions remain qualitatively unchanged (see Supplementary Note 5 and Figs. S9-S14) and we therefore only present the results from the variable rates models in the main text.

### Size-dependency in the brain-body relationship across species
To determine the size dependency effect on brain and body size slopes, we ran two additional PGLS models in the same way described above. These models estimated a separate intercept and slope for each class (class slope model) or each subclade (subclade slope model) respectively. We then estimated a relationship between the median slope value and median body size for each group (class or subclade) using maximum likelihood PGLS models implemented in *caper*[30], estimating phylogenetic signal using Lambda[85].

### Analyses incorporating within-species variation
To investigate the possibility that heterogeneity in the brain and body size relationship observed amongst individual species could give rise to the observed curvature across species, we conducted a series of analyses using phylogenetic generalized linear mixed models (PGLMMs) implemented in MCMCglmm[33]. In these models, we incorporate phylogenetic information as a random effect using the phylogenetic variance-covariance matrix. Species-level variation is incorporated by the inclusion of all values for species with at least 10 individual measurements and an additional random effect allocating each point to a given species.

Prior to our PGLMMs, we first ran a reduced global curve model, replicating our global curve model using only the sample of species for which we had within-species data, and the brain and body size measurements as reported in Supplementary Data 1, ensuring that we still recovered significant curvature in this reduced dataset. This reduced sample of taxa still displays significant negative curvature (mean $\beta$ = -0.04, $p_x$ = 0.02). We then ran a global curve with species-level variation PGLMM on the same reduced sample of species but this time including all individual measurements. This model included both body size and a second order quadratic term as fixed effects along with the phylogeny and species association as random effects. Phylogenetic signal was assessed using heritability. For all continuous fixed effects, we used a normally distributed diffuse prior as in the default options[33]. For random effects we used non-informative parameter-expanded priors ($V$ = 1, nu = 1, alpha.mu = 0, alpha.V = 1000).

We then used within-group centering[32] to separate body size into two components: across-species body size (a species-level average of body size) and within-species body size (calculated as the difference between each species and the species-level mean). In doing so, it becomes possible to estimate both the brain and body size relationship (including curvature) across all species alongside an average effect of within-species allometric variability – within a single statistical model. We refer to this as our *hierarchical model*, which includes three fixed effects (across-species body size and its second order quadratic term as well as within-species body size) and three random effects (the phylogenetic variance-covariance matrix, a species identifier, and within-species slopes). Including species-level slopes as a random effect allows us to incorporate the variability in slope observed across individual species[4].

It is not currently possible to estimate rate heterogeneity in a PGLMM (see above); all PGLMM analyses were therefore performed on the time tree of life limited to the species included within the analysis, where branch lengths were proportional to time. However, our conclusions remained qualitatively identical when we use a tree where branch lengths have been scaled to reflect the median rate of evolution given the *global curve* model (i.e. branch lengths = $r \times t$).

### Reporting summary
Further information on research design is available in the Nature Portfolio Reporting Summary linked to this article.

## Data availability
All data analysed in this study are available in the main text or the supplementary materials, along with the original sources from which they were obtained (Supplementary Data 1). Source data are provided with this paper.

## Code availability
We provide code developed as a part of this work in the supplementary material (Supplementary Code 1), but all other analyses were performed using publicly available, published and peer-reviewed programs which are cited appropriately in the text.

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

## Acknowledgements
We would like to thank the University of Reading Evolutionary Biology Research Group and George Butler for helpful discussions about this work. CV was funded by a Leverhulme Trust Leadership Award RL-2019-012.

## Author contributions
JB, RB, and CV conceptualized the study and developed the methodology. JB performed the investigation under the supervision of CV. JB wrote the original draft and generated the visualizations. JB, RB, and CV were responsible for reviewing and editing successive drafts of the manuscript.

## Competing interests
The authors declare no competing interests.
