## [Transparent Peer Review file · Nature Communications]

Macroevolutionary brain scaling is a microevolutionary metaphenomenon

Corresponding Author: Professor Chris Venditti

Version 0:

Reviewer comments:

Reviewer #1

(Remarks to the Author)

General comments:

The authors show that the scaling of brain size is curvilinear across the tree of animal life. They further explain this curvilinearity as being the result of taxonomic differences in the scaling exponent for brain size that become less steep with increasing mean body size. This is a notable, well-executed contribution, but I have two major concerns that I feel that the authors should address.

1) It may be worthwhile to let readers know that curvilinearity (or other forms of non-linearity) of body-size scaling occurs for other traits, not only across species (e.g., Zeuthen 1953; Hayssen & Lacy 1985; Kolokotronis et al. 2010; Ehnes et al. 2011; Bueno & López-Urrutia 2014; Douhard et al. 2016; Glazier 2024a) but also within species (Glazier 2005, 2024b). The authors mention that curvilinearity occurs for metabolic scaling (L 213-215; also see Mori et al. 2010), but not for other traits – e.g., ingestion rate (Bueno & López-Urrutia 2014; Douhard et al. 2016); lifespan (Bueno & López-Urrutia 2014); productivity (Bueno & López-Urrutia 2014); and offspring size and number (Glazier 2024a).

2) The authors' explanation of the overall curvilinear scaling of brain size in animals partially resembles the view that metabolic scaling across mammal species is affected by scaling within species (Heusner 1982, 1991), though this view emphasizes the effects of varying intraspecific intercepts rather than slopes on the interspecific relationship. Other investigators have also linked curvilinear metabolic scaling of mammals to taxonomic differences in scaling exponents (e.g., Hayssen & Lacy 1985; Clarke et al. 2010; Ehnes et al. 2011). Indeed, Ehnes et al. (2011) showed that curvilinear metabolic scaling of terrestrial invertebrates disappeared after phylogenetic groups were included in a complex scaling model. I mention this merely to show that the authors are not the first to attempt to explain broad biological scaling patterns in terms of narrower taxonomic differences in scaling. It would be useful to mention that the authors examined the effects of both within and across species scaling relationships on the overall scaling of brain size and show that only within species relationships can explain the curvature, a unique and interesting finding!

Specific comments:

L 37-38: This is a misleading statement that partially contradicts the authors' claim that taxonomic differences cause the overall curvilinear relationship. Please clarify. I suggest not using the problematic word "dissolves".

L 94: Do you really mean 54g for the brain mass of a whale?

L 98-99: Figures 2B, 2C & 3 show that large-bodied groups show less (not more) change in brain mass with increasing body mass (the scaling slope is lower). Please correct.

L 105, 118, etc.: In the scaling literature, the scaling slope is usually called an "exponent", whereas the scaling intercept is called a "coefficient". Using the term coefficient for slope may be confusing to some readers.

L 183: Insert "to" between "reported" and "be".

Literature cited:

Bueno, J., & López-Urrutia, Á. (2014). Scaling up the curvature of mammalian metabolism. *Frontiers in Ecology and Evolution*, 2, 61.

Clarke, A., Rothery, P., & Isaac, N. J. (2010). Scaling of basal metabolic rate with body mass and temperature in mammals. *Journal of Animal Ecology*, 79(3), 610-619.

Douhard, F., Lemaître, J. F., Rauw, W. M., & Friggens, N. C. (2016). Allometric scaling of the elevation of maternal energy intake during lactation. *Frontiers in zoology*, 13, 1-12.

Ehnes, R. B., Rall, B. C., & Brose, U. (2011). Phylogenetic grouping, curvature and metabolic scaling in terrestrial invertebrates. *Ecology Letters*, 14(10), 993-1000.

Glazier, D. S. (2005). Beyond the '3/4-power law': variation in the intra-and interspecific scaling of metabolic rate in animals. *Biological Reviews*, 80(4), 611-662.

Glazier, D. S. (2024a). Age-specific mortality predicts body-mass scaling of offspring mass and number. *Evolutionary Ecology*, 38(4), 513-535.

Glazier, D. S. (2024b). Multiphasic allometry: the reality and significance of ontogenetic shifts in the body-mass scaling of metabolic rate. *Academia Biology*, 2(4).

Hayssen, V., & Lacy, R. C. (1985). Basal metabolic rates in mammals: taxonomic differences in the allometry of BMR and body mass. *Comparative Biochemistry and Physiology Part A: Physiology*, 81(4), 741-754.

Heusner, A. A. (1982). Energy metabolism and body size I. Is the 0.75 mass exponent of Kleiber's equation a statistical artifact? *Respiration Physiology*, 48(1), 1-12.

Heusner, A. A. (1991). Size and power in mammals. *Journal of Experimental Biology*, 160(1), 25-54.

Kolokotronis, T., Savage, V., Deeds, E. J., & Fontana, W. (2010). Curvature in metabolic scaling. *Nature*, 464(7289), 753-756.

Mori, S., Yamaji, K., Ishida, A., Prokushkin, S.G., Masyagina, O.V., Hagihara, A., Hoque, A.R., Suwa, R., Osawa, A., Nishizono, T. and Ueda, T., 2010. Mixed-power scaling of whole-plant respiration from seedlings to giant trees. *Proceedings of the National Academy of Sciences*, 107(4), pp.1447-1451.

Zeuthen, E. (1953). Oxygen uptake as related to body size in organisms. *Quarterly Review of Biology*, 28(1), 1-12.

Douglas S. Glazier

(Remarks on code availability)

I did not attempt to run the code. Sorry!

Reviewer #2

(Remarks to the Author)

I generally like this paper. It is a followup to their mammal study.

My main question is why? The authors show clear statistical patterns, but no theory to explain them. For example, the metabolic theory proposed underlying fractal like infrastructure for optimal resource distribution. Do the authors have a theoretical explanation that would explain/give rise to the observed patterns?

(Remarks on code availability)

Will be provided

Reviewer #3

(Remarks to the Author)

This study includes an impressive amount of data from vertebrates and uses relatively new phylogenetic comparative methods to understand the relationship between brain and body size. The authors find that, rather than a linear allometric relationship between these variables, there is a curvilinear relationship. This implies, if I am interpreting their analyses correctly, that larger bodied animals have a lower slope to their BBM relationship than smaller animals. The authors report that no major vertebrate groups deviate from this "universal" formula, yet they do find deviations in four less inclusive vertebrate groups. First, I do think it is a mistake to dismiss these as minor curiosities since they include rather large clades. Some explanation of their deviation would be nice. My main concern, however, is that it sounds like the authors are arguing

for a purely structuralist paradigm, and one in which brain size evolution cannot occur apart from an increase in body size. This is a fairly extraordinary claim. Even if there is found to be a universal curvilinear pattern, this does not exclude the possibility that individual species might not deviate from that pattern and have higher than expected brain sizes for a given body size due to ecological/behavioral variables. I think most researchers recognize that brain and body size are linked at some level (even a linear allometric relationship implies this), but it is the pattern that is interesting here.

The authors mention that most vertebrate clades and subclades adhere to this universal pattern, but the text and figures make it seem that it is really a phenomenon of large-bodied animals. What would be interesting is to take a group of small-bodied animals such as birds (one for which a genomic tree is available; Jarvis et al., 2015) to see if the same pattern is present within a more confined set of data to test for significant deviations from this pattern. Is there a size threshold at which the curvilinearity becomes pronounced? Does it differ from other studies of BBM in this group (Ksepka et al., 2021)?

For the intraspecific study, the authors mention that they “extracted brain and body size data adjusted for sex and age”. It’s not clear what they mean by this. How did they adjust the data? Were juvenile individuals included within the analysis? If so, does that mean we are looking at growth curves as opposed to intraspecific variation?

On pg. 6, the sentence in line 183 that begins “In line with this...” and spans to line 185 is confusing. It’s unclear what they are trying to say here.

The next sentence of page 6 cites the Kverková paper, which is about neuron numbers and not connectivity.

On page 6, lines 203-204: While I agree with the sentence that begins “Instead, we need to understand...”, I do not think that this paper addresses process. It is simply describing intra- and interspecific patterns.

Another concern that I have is that the tree comes from timetree.org, which means that it is not the product of a phylogenetic analysis (or even a concatenated tree of other analyses) but rather just a summary tree to be used for divergence dates. This is the first time that I’ve seen someone use the website in this way. I am unsure how it might affect the analysis, but it would be helpful if the authors provided some justification for using this tree. In the methods, they also refer to a “chordate level” tree. I’m not sure what this means—a tree of Chordata?

I also would like to see the author’s drop non-avian reptiles as a grouping in their analysis since this is a paraphyletic group and not a clade. It would be better to just include the various groups of reptiles (ie., squamates, turtles, crocs, and birds).

In the methods section, in the “Within-Species Data” section, it looks like the wrong Tsuboi et al. paper was cited.

(Remarks on code availability)

Version 1:

Reviewer comments:

Reviewer #1

(Remarks to the Author)

The authors have satisfactorily addressed all my comments. Good job!

Reviewer #2

(Remarks to the Author)

Reviewer #3

(Remarks to the Author)

The authors have done a thorough job of addressing all of my concerns, and I would suggest publication.

REVIEWER COMMENTS

Reviewer #1 (Remarks to the Author):

General comments:

The authors show that the scaling of brain size is curvilinear across the tree of animal life. They further explain this curvilinearity as being the result of taxonomic differences in the scaling exponent for brain size that become less steep with increasing mean body size. This is a notable, well-executed contribution, but I have two major concerns that I feel that the authors should address.

We thank Professor Glazier for his generally positive assessment of our paper. We address each of his concerns in the following responses.

1) It may be worthwhile to let readers know that curvilinearity (or other forms of non-linearity) of body-size scaling occurs for other traits, not only across species (e.g., Zeuthen 1953; Hayssen & Lacy 1985; Kolokotronis et al. 2010; Ehnes et al. 2011; Bueno & López-Urrutia 2014; Douhard et al. 2016; Glazier 2024a) but also within species (Glazier 2005, 2024b). The authors mention that curvilinearity occurs for metabolic scaling (L 213-215; also see Mori et al. 2010), but not for other traits – e.g., ingestion rate (Bueno & López-Urrutia 2014; Douhard et al. 2016); lifespan (Bueno & López-Urrutia 2014); productivity (Bueno & López-Urrutia 2014); and offspring size and number (Glazier 2024a).

We agree that this is a worthwhile inclusion to our manuscript. We have therefore now incorporated acknowledgement of non-linearity and curvilinearity of other traits right up front in the first paragraph of our introduction (see L36-40). In addition, we have now re-written our concluding paragraph to explicitly incorporate the established curvilinearity between metabolic rate and body size and how this has led to predictions of curvature in other traits (Bueno & López-Urrutia 2014). We use this section to lead into some new text that highlights other traits that have documented curvature and non-linearity and what this might mean in the context of within-species heterogeneity. This new section can be found in the last paragraph of the manuscript's main text (from L218 -).

In the earlier version of the discussion / concluding paragraph mentioned above, we did already cite Glazier's 2024a paper discussing ontogenetic variation within individual species. However, we have now expanded this in accordance with the above and added the additional citation recommended here.

All citations recommended by Professor Glazier have been incorporated into the new revision – along with several other supporting references.

2) The authors' explanation of the overall curvilinear scaling of brain size in animals partially resembles the view that metabolic scaling across mammal species is affected by scaling within species (Heusner 1982, 1991), though this view emphasizes the effects of varying intraspecific intercepts rather than slopes on the interspecific relationship. Other investigators have also linked curvilinear metabolic scaling of mammals to taxonomic differences in scaling exponents (e.g., Hayssen & Lacy 1985; Clarke et al. 2010; Ehnes et al. 2011). Indeed, Ehnes et al. (2011) showed that curvilinear metabolic scaling of terrestrial invertebrates disappeared after phylogenetic groups were included in a complex scaling model. I mention this merely to show that the authors are not the first to attempt to explain broad biological scaling patterns in terms of narrower taxonomic differences in scaling. It

would be useful to mention that the authors examined the effects of both within and across species scaling relationships on the overall scaling of brain size and show that only within species relationships can explain the curvature, a unique and interesting finding!

We appreciate the positive comment. We agree that recognizing earlier contributions as outlined here is important and places our novel results in a more complete context. We have therefore made several changes to the manuscript to explicitly incorporate these earlier views.

Firstly, we have incorporated Heusner's suggested association into the section of our manuscript where we introduce the idea of looking at variation in slopes amongst species (L156-162) – emphasizing our expectations for variability in within-species allometries.

Secondly, we have modified the discussion paragraph regarding within-species relationships in the same section (L218-224). In this edited paragraph, we incorporate the recommended references associated with exploring curvilinearity in association with lower-level taxonomic variation, but additionally emphasize our novel results as suggested.

Specific comments:

L 37-38: This is a misleading statement that partially contradicts the authors' claim that taxonomic differences cause the overall curvilinear relationship. Please clarify. I suggest not using the problematic word "dissolves".

We have modified this statement in order to clarify and have omitted the word "dissolves" from the new version.

L 94: Do you really mean 54g for the brain mass of a whale?

We apologize for this error – we have amended the value to 6.9kg in the new version (L120-123).

L 98-99: Figures 2B, 2C & 3 show that large-bodied groups show less (not more) change in brain mass with increasing body mass (the scaling slope is lower). Please correct.

We have now amended this error.

L 105, 118, etc.: In the scaling literature, the scaling slope is usually called an "exponent", whereas the scaling intercept is called a "coefficient". Using the term coefficient for slope may be confusing to some readers.

We recognize this could be confusing. We retain the original phrase "evolutionary regression coefficient" where we introduce in the context of its original citation (L 121) but have added an additional clarification that this refers to the slope parameter. We also clarify this when it is used again later on in the manuscript.

Elsewhere, we have amended the use of the word coefficient to explicitly refer to the relevant parameters. For example, in our figures (and captions) we amend "slope coefficients" to "slope parameters". We have made the same adjustments throughout our supplementary material.

L 183: Insert "to" between "reported" and "be".

We have amended this typographical error in the new version of our manuscript.

Literature cited:

- Bueno, J., & López-Urrutia, Á. (2014). Scaling up the curvature of mammalian metabolism. *Frontiers in Ecology and Evolution*, 2, 61.
- Clarke, A., Rothery, P., & Isaac, N. J. (2010). Scaling of basal metabolic rate with body mass and temperature in mammals. *Journal of Animal Ecology*, 79(3), 610-619.
- Douhard, F., Lemaître, J. F., Rauw, W. M., & Friggens, N. C. (2016). Allometric scaling of the elevation of maternal energy intake during lactation. *Frontiers in zoology*, 13, 1-12.
- Ehnes, R. B., Rall, B. C., & Brose, U. (2011). Phylogenetic grouping, curvature and metabolic scaling in terrestrial invertebrates. *Ecology Letters*, 14(10), 993-1000.
- Glazier, D. S. (2005). Beyond the '3/4-power law': variation in the intra-and interspecific scaling of metabolic rate in animals. *Biological Reviews*, 80(4), 611-662.
- Glazier, D. S. (2024a). Age-specific mortality predicts body-mass scaling of offspring mass and number. *Evolutionary Ecology*, 38(4), 513-535.
- Glazier, D. S. (2024b). Multiphasic allometry: the reality and significance of ontogenetic shifts in the body-mass scaling of metabolic rate. *Academia Biology*, 2(4).
- Hayssen, V., & Lacy, R. C. (1985). Basal metabolic rates in mammals: taxonomic differences in the allometry of BMR and body mass. *Comparative Biochemistry and Physiology Part A: Physiology*, 81(4), 741-754.
- Heusner, A. A. (1982). Energy metabolism and body size I. Is the 0.75 mass exponent of Kleiber's equation a statistical artifact? *Respiration Physiology*, 48(1), 1-12.
- Heusner, A. A. (1991). Size and power in mammals. *Journal of Experimental Biology*, 160(1), 25-54.
- Kolokotronis, T., Savage, V., Deeds, E. J., & Fontana, W. (2010). Curvature in metabolic scaling. *Nature*, 464(7289), 753-756.
- Mori, S., Yamaji, K., Ishida, A., Prokushkin, S.G., Masyagina, O.V., Hagihara, A., Hoque, A.R., Suwa, R., Osawa, A., Nishizono, T. and Ueda, T., 2010. Mixed-power scaling of whole-plant respiration from seedlings to giant trees. *Proceedings of the National Academy of Sciences*, 107(4), pp.1447-1451.
- Zeuthen, E. (1953). Oxygen uptake as related to body size in organisms. *Quarterly Review of Biology*, 28(1), 1-12.

Douglas S. Glazier

Reviewer #1 (Remarks on code availability):

I did not attempt to run the code. Sorry!

Reviewer #2 (Remarks to the Author):

I generally like this paper. It is a followup to their mammal study.

We thank the reviewer for the positive comment on our manuscript.

My main question is why? The authors show clear statistical patterns, but no theory to

explain them. For example, the metabolic theory proposed underlying fractal like infrastructure for optimal resource distribution. Do the authors have a theoretical explanation that would explain/give rise to the observed patterns?

We do not know the theoretical explanation for the patterns we observe – this is for future work. We do, however, provide a mechanistic explanation in terms of the within-species variation. We additionally test several other factors that might explain the size-dependency in the manuscript but find that none do. What remains to be seen is what exactly drives the variation in allometries within individual species – for this, we point readers towards individual-level dynamics and ontogenetic variation.

We explicitly state this in the concluding paragraphs of the manuscript, and we hope that our paper might inspire a path of discovery from which empirical or biological explanations might arise.

Reviewer #2 (Remarks on code availability):

Will be provided

Reviewer #3 (Remarks to the Author):

This study includes an impressive amount of data from vertebrates and uses relatively new phylogenetic comparative methods to understand the relationship between brain and body size. The authors find that, rather than a linear allometric relationship between these variables, there is a curvilinear relationship. This implies, if I am interpreting their analyses correctly, that larger bodied animals have a lower slope to their BBM relationship than smaller animals.

We thank the reviewer for the generally positive summary of our work.

The authors report that no major vertebrate groups deviate from this “universal” formula, yet they do find deviations in four less inclusive vertebrate groups. First, I do think it is a mistake to dismiss these as minor curiosities since they include rather large clades. Some explanation of their deviation would be nice.

We identify deviations from global curvature in five groups: salamanders (n = 55), Atlantogenata (n = 47), Labriformes (n = 92), Falconiformes, n = 26, and Charadriiformes (n = 140). Please note that in our revised manuscript, we repeated our analyses recoding a couple of mis-classified bird species (N = 3). In these corrected analyses, we identified an additional deviation (Falconiformes). None of these are particularly large clades with respect to the total number of observations studied here (each representing <3% of total species).

Notably, four of these five clades do not differ from the main pattern observed across all other taxa in that they exhibit negative curvature. However, we did not intend to give the impression of dismissing these groups as minor curiosities. Previously, we explicitly explored the deviation observed within each of these clades in some detail in our supplementary material. We have now substantially expanded this section of our supplementary material – in section titled “Subgroups identified as different from global curvature”. This includes the incorporation of several new figures that more clearly outline the deviations (Figures S1-S6).

To emphasize the importance of these potential differences from the global curvature, we also now include some of this exploration in our main text. Firstly, in L102-106 we demonstrate that the difference in Atlantogenata is caused by just 5 taxa (2 elephants and 3 sirenians). We then go onto highlight the fact that Charadriiformes is left as the uniquely positive deviation – with a positive curvature existing for all three major suborders within the group (Charadrii, n = 35; Scolopaci, n = 47; Lari, n = 58). What is causing the difference in these seabirds is an interesting question and yet unknown. We now highlight and suggest potential causes for this curiosity such as sexual dimorphism and mating system diversity in our manuscript (L106-111).

Finally, we have additionally amended the text to no longer refer to “minor curiosities” (L114), as well as referring explicitly to the extensive exploration within the supplementary materials in the main text (L100-108).

My main concern, however, is that it sounds like the authors are arguing for a purely structuralist paradigm, and one in which brain size evolution cannot occur apart from an increase in body size. This is a fairly extraordinary claim. Even if there is found to be a universal curvilinear pattern, this does not exclude the possibility that individual species might not deviate from that pattern and have higher than expected brain sizes for a given body size due to ecological/behavioral variables. I think most researchers recognize that

brain and body size are linked at some level (even a linear allometric relationship implies this), but it is the pattern that is interesting here.

We are not arguing for this, and we agree that the pattern is by and large the most exciting part of our results. We agree that individual species can deviate from the relationship we describe – not only do we present variability in the form of species-level slopes, but we explicitly incorporate and detect this variability as a component of our model in the form of variable rates of morphological evolution. We include these results in the main text (L131-16) and additionally include a supplementary figure (Figure S7), several supplementary tables (Tables S3-S9), and additional supplementary text that incorporates these results (section titled ‘Summary of rate heterogeneity’). We hope that this clarifies our argument.

The authors mention that most vertebrate clades and subclades adhere to this universal pattern, but the text and figures make it seem that it is really a phenomenon of large-bodied animals. What would be interesting is to take a group of small-bodied animals such as birds (one for which a genomic tree is available; Jarvis et al., 2015) to see if the same pattern is present within a more confined set of data to test for significant deviations from this pattern. Is there a size threshold at which the curvilinearity becomes pronounced? Does it differ from other studies of BBM in this group (Ksepka et al., 2021)?

We think that there is some misunderstanding of our paper and analyses here.

Firstly, this pattern is not restricted to large-bodied animals. It is possible that this comment is linked to the typo noted above by Reviewer #1 in that we erroneously state that large-bodied animals have proportionally more change in brain mass. We have amended this in the new version to read, correctly, that large-bodied animals have proportionally less change. What this means is simply that there is a size dependency in the BBM relationship. However, we also demonstrate that all major animal clades – which notably includes the particularly small-bodied insects – demonstrate the same pattern. Even within vertebrates (which the reviewer seems to focus in on throughout their report), we find the same pattern in many relatively small-bodied clades – and sub-clades. This includes groups like frogs, gobies, seahorses and pipefish, bats, and rodents.

Secondly, we already include birds in our study. We find not only do birds as a whole not deviate from the global pattern (Fig. 1) but even within birds, it is only one group of seabirds (see above) that exhibits any difference.

Thirdly, it is not possible to identify a size threshold at which the curvilinearity becomes “pronounced”, because our results imply a uniform curvilinear relationship acting across all species and taxa that is explicitly linked to size and is documented across the entire size range of our sample.

Finally, our results differ from other studies of BBM in various groups – including Ksepka’s study of birds mentioned by the reviewer – in that ours is thus far the only study to incorporate curvilinearity in place of taxonomic or clade-level heterogeneity in BBM allometric slopes. Our results show why their linear slope parameters vary (they are size-dependent phenomena which disappear when we apply our model). We cite previous work in this context – including Ksepka’s study – in various places throughout our manuscript.

For the intraspecific study, the authors mention that they “extracted brain and body size data adjusted for sex and age”. It’s not clear what they mean by this. How did they adjust the data? Were juvenile individuals included within the analysis? If so, does that mean we are looking at growth curves as opposed to intraspecific variation?

We thank the reviewer for noticing this – there was a minor typographical error here in that the data was adjusted for sex and method rather than sex and age. No juveniles were included. We have corrected this error in the new version of the manuscript (L306-308). In our original manuscript, we stated that we use the data and protocol associated with the original paper (Tsuboi et al 2018, NEE). The original ‘adjustments’ were simply to centre group means with respect to sex and measurement methods. We have added some description of this process to the new version of the manuscript (L306-308), but the full protocol and code are available from the original paper.

On pg. 6, the sentence in line 183 that begins “In line with this...” and spans to line 185 is confusing. It’s unclear what they are trying to say here.

We have simplified this sentence to make clearer our intended meaning – which is simply that various factors associated with brain size on a macro-evolutionary scale do not explain the size dependence (L226-228).

The next sentence of page 6 cites the Kverková paper, which is about neuron numbers and not connectivity.

We have amended this to read neuron number (l229).

On page 6, lines 203-204: While I agree with the sentence that begins “Instead, we need to understand...”, I do not think that this paper addresses process. It is simply describing intra- and interspecific patterns.

We disagree that we are just simply describing variation, rather we are showing how one (intra-specific variation) affects the other (inter-specific variation) – and this is the point we are trying to make. We have now amended this sentence to clarify that we are drawing attention to the need to study how intra-specific and inter-specific evolutionary processes interact.

Another concern that I have is that the tree comes from timetree.org, which means that it is not the product of a phylogenetic analysis (or even a concatenated tree of other analyses) but rather just a summary tree to be used for divergence dates. This is the first time that I’ve seen someone use the website in this way. I am unsure how it might affect the analysis, but it would be helpful if the authors provided some justification for using this tree.

Trees obtained from timetree.org are summarized super-trees consolidated from multiple sources of published time-scaled phylogenies (Hedges 2015, Kumar 2022). Whilst the website is designed to be used to extract divergence dates between various taxa, there are various tools available that are designed explicitly for extracting single trees – which represent a good summary of the current understanding of species relationships and divergences.

- **Hedges SB, Marin J, Suleski M, Paymer M, Kumar S. Tree of life reveals clock-like speciation and diversification. Molecular biology and evolution. 2015;32(4):835-45).**

- Kumar S, Suleski M, Craig JM, Kasprowicz AE, Sanderford M, Li M, et al. TimeTree 5: An Expanded Resource for Species Divergence Times. *Molecular Biology and Evolution*. 2022;39(8):msac174.

The use of the time tree of life from this resource for phylogenetic comparative analysis is not considered controversial, nor is it novel to this study – a cursory search of the literature reveals dozens of studies have used trees available from this resource in a similar way in only the last year (see below for just a few examples).

- de Brito Freire-Jr, Geraldo, et al. "Fostering Biodiversity in Neotropical Savannas: Fire as a Diversity Driver for Fruit-Feeding Butterfly Assemblages in the Cerrado." *Austral Ecology* 50.3 (2025): e70053.
- Glick, Lior, et al. "Phylogenetic Analysis of 590 Species Reveals Distinct Evolutionary Patterns of Intron–Exon Gene Structures Across Eukaryotic Lineages." *Molecular Biology and Evolution* 41.12 (2024): msae248.
- Hermanson, Guilherme, and Serjoscha W. Evers. "Shell Constraints on Evolutionary Body Size–Limb Size Allometry Can Explain Morphological Conservatism in the Turtle Body Plan." *Ecology and Evolution* 14.11 (2024): e70504.
- Mallik, Rittika, et al. "Investigating the Impact of Whole-Genome Duplication on Transposable Element Evolution in Teleost Fishes." *Genome Biology and Evolution* 17.1 (2025): evae272.
- Matsuda, Yuki, and Takashi Makino. "Comparative genomics reveals convergent signals associated with the high metabolism and longevity in birds and bats." *Proceedings of the Royal Society B* 291.2029 (2024): 20241068.
- Matthews, Sophie, et al. "Variable gene copy number in cancer-related pathways is associated with cancer prevalence across mammals." *Molecular Biology and Evolution* 42.3 (2025): msaf056.
- Pagel, Mark, and Andrew Meade. "Trait macroevolution in the presence of covariates." *Nature Communications* 16.1 (2025): 4555.
- Policarpo, M., Baldwin, M.W., Casane, D. et al. Diversity and evolution of the vertebrate chemoreceptor gene repertoire. *Nat Commun* 15, 1421 (2024).
- Roberts, Miles D., and Emily B. Josephs. "k-mer-based diversity scales with population size proxies more than nucleotide diversity in a meta-analysis of 98 plant species." *Evolution Letters* (2025): qraf011.
- Szasz-Green, Taylor, et al. "Comparative phylogenetics reveal clade-specific drivers of recombination rate evolution across vertebrates." *Molecular Biology and Evolution* 42.5 (2025): msaf100.
- Wilson, Lauren N., et al. "Global latitudinal gradients and the evolution of body size in dinosaurs and mammals." *Nature Communications* 15.1 (2024): 2864.
- Winkler, Lennart, et al. "Pre-Copulatory Sexual Selection Predicts Sexual Size Dimorphism: A Meta-Analysis of Comparative Studies." *Ecology Letters* 27.9 (2024): e14515.
- Yu Dan and Wiens John J. 2024The causes of species richness patterns among clades *Proc. R. Soc. B*.29120232436

In the methods, they also refer to a “chordate level” tree. I’m not sure what this means—a tree of Chordata?

Yes, that is what it means. We have amended for clarity.

I also would like to see the author’s drop non-avian reptiles as a grouping in their analysis

since this is a paraphyletic group and not a clade. It would be better to just include the various groups of reptiles (ie., squamates, turtles, crocs, and birds).

We recognize that this is a problematic grouping – it was included mostly in order to incorporate data from crocodylians and testudines. Whilst we retain our original analyses across the complete dataset, we have additionally repeated our clade-level analyses removing crocodylians and analysing squamates, birds, and testudines as separate clades – with identical conclusions. We have integrated this into the new version of the manuscript – including a new version of Figure 1.

Importantly, whilst our first clade-level model studies non-avian reptiles as a group, in all subsequent models we consider lower-level taxonomy. In these analyses, only species from squamates and birds are included owing to the paucity of data in the other non-avian reptiles. Our results from within squamates and birds reveal the same patterns as those from across the broader clades.

In the methods section, in the “Within-Species Data” section, it looks like the wrong Tsuboi et al. paper was cited.

We thank the reviewer for noticing this error. It has been amended in the new version.